# Targeting of the Cancer-Associated Fibroblast—T-Cell Axis in Solid Malignancies

**DOI:** 10.3390/jcm8111989

**Published:** 2019-11-15

**Authors:** Tom J. Harryvan, Els M. E. Verdegaal, James C. H. Hardwick, Lukas J. A. C. Hawinkels, Sjoerd H. van der Burg

**Affiliations:** 1Department of Gastroenterology & Hepatology, Leiden University Medical Center, 2333 ZA Leiden, The Netherlands; t.j.harrijvan@lumc.nl (T.J.H.); J.C.H.Hardwick@lumc.nl (J.C.H.H.); L.J.A.C.Hawinkels@lumc.nl (L.J.A.C.H.); 2Department of Medical Oncology, Oncode Institute, Leiden University Medical Center, 2333 ZA Leiden, The Netherlands; E.M.E.Verdegaal@lumc.nl

**Keywords:** cancer-associated fibroblast, tumor immunology, T-cell based immunotherapy

## Abstract

The introduction of a wide range of immunotherapies in clinical practice has revolutionized the treatment of cancer in the last decade. The majority of these therapeutic modalities are centered on reinvigorating a tumor-reactive cytotoxic T-cell response. While impressive clinical successes are obtained, the majority of cancer patients still fail to show a clinical response, despite the fact that their tumors express antigens that can be recognized by the immune system. This is due to a series of other cellular actors, present in or attracted towards the tumor microenvironment, including regulatory T-cells, myeloid-derived suppressor cells and cancer-associated fibroblasts (CAFs). As the main cellular constituent of the tumor-associated stroma, CAFs form a heterogeneous group of cells which can drive cancer cell invasion but can also impair the migration and activation of T-cells through direct and indirect mechanisms. This singles CAFs out as an important next target for further optimization of T-cell based immunotherapies. Here, we review the recent literature on the role of CAFs in orchestrating T-cell activation and migration within the tumor microenvironment and discuss potential avenues for targeting the interactions between fibroblasts and T-cells.

## 1. Introduction

The notion that the tumor stroma is an important factor in determining patient prognosis and survival has now found a firm base in a multitude of solid tumors [1,2,3,4,5]. Tumors with high stromal content correlate with an increased risk of distant metastases and worse overall patient survival [6,7]. Further stratification of the different cellular components that comprise the tumor stroma, including endothelial cells, immune cells and CAFs, has pointed towards a prominent role of CAFs in contributing to this dismal prognosis [1,8].

As the major constituent of the tumor stroma, CAFs are a distinct cellular entity exhibiting mesenchymal features, reflected by their lack of expression of markers of either endothelial, epithelial or immune origin. Moreover, CAFs are characterized by their spindle-shaped morphology and the expression of certain fibroblast activation markers, including alpha-smooth muscle actin (αSMA) and fibroblast-activation protein (FAP). The expression of these molecules is upregulated in most activated fibroblasts, which occurs during wound healing processes and in solid tumors. Since CAFs share many similarities to wound-healing associated fibroblasts, tumors have been considered as a ‘wound that does not heal’, leading to perpetual activation of resident fibroblasts [9,10]. Originally, CAFs were reported as one single cell population derived from cells of different origins. However, more recently, specific subsets of CAFs have been identified based on the expression of other membranous and secreted proteins, including platelet-derived growth factor receptors alpha and beta (PDGF-Rα, PDGF-Rβ), periostin (POSTN), tenascin C (TN-C), podoplanin (PDPN) and endoglin. Although this provides valuable information, a comprehensive characterization of the expression of these markers on CAFs and their distinct roles in tumor progression has remained challenging due to the enormous heterogeneity of these cells and the analyses performed [11,12,13,14,15].

CAF heterogeneity might be partially explained by the fact that fibroblasts within one tumor can originate from different cellular precursors and from distinct cellular locations. First, resident fibroblasts can adopt a CAF phenotype in response to factors secreted in the TME, such as Transforming Growth Factor Beta (TGF-β), Wnt, PDGF and interleukins (Figure 1A) [16,17,18,19,20,21]. Secondly, both endothelial and epithelial cells within the TME can adopt a more mesenchymal CAF-like phenotype, also largely driven by TGF-β signaling, a process termed endothelial-to-mesenchymal transition (EndoMT) and epithelial-to-mesenchymal transition (EMT), respectively (Figure 1B,C) [22,23,24]. Thirdly, bone-marrow derived mesenchymal stem cells (MSCs) can be recruited into the tumor and adopt a CAF-like phenotype upon activation by various cytokines in the TME (Figure 1D) [25,26,27]. Lastly, transdifferentiation of pericytes or smooth muscle cells can also give rise to a CAF-like phenotype (Figure 1E) [9,28]. The final product of all these differential routes leads to a mesenchymal-like cell characterized by high motility, proliferation and an enhanced secretory phenotype capable of promoting cancer progression through stimulation of angiogenesis, tumor cell proliferation, invasion and extravasation, remodeling of the extracellular matrix (ECM) and acquisition of chemotherapy resistance (Figure 1F) [9,29]. Finally, CAFs have been shown to play a critical role in the regulation of anti-tumor immunity.

With regard to the regulation of anti-tumor immunity, a growing body of evidence supports the view that CAFs predominantly display a pro-tumorigenic role, although anti-tumorigenic CAF subsets have also been described [30,31]. Recent studies have focused on identifying CAF subsets in order to better understand their function in the TME. A general overview of the interactions between CAFs and cells of the immune system has been provided elsewhere [32,33].

In this review, we exclusively focus on the interactions between CAFs and T-cells since the majority of current immunotherapeutic modalities center around eliciting or reinvigorating a tumor-specific T cell response, either through checkpoint inhibition, therapeutic vaccination, adoptive transfer of ex vivo expanded tumor-specific (TCR-transgenic or chimeric antigen receptor (CAR)) T-cells, oncolytic virus therapy or a combination of these strategies together or with the immunomodulatory properties of chemotherapy/radiation. Finally, we will also discuss avenues to exploit the knowledge of these interactions for the rational improvement of T-cell-based immunotherapy.

## 2. Cancer-Associated Fibroblast Phenotypes

### 2.1. Fibroblast Subtyping in Solid Malignancies Identifies a Pan-Tumor Inflammatory CAF Subset

The inherent difficulty in studying fibroblast—T-cell interactions is the lack of understanding of the molecular and functional heterogeneity of CAFs within the TME. Currently, there is a lack of consensus on specific markers to differentiate normal fibroblasts from CAFs. Traditionally, both normal fibroblasts and CAFs have been defined by the absence of markers specific for cells of either endothelial (CD31), epithelial (EpCam) or immune origin (CD45), as well as by the expression of a combination of a relatively fixed set of markers, including CD90 (Thy-1), αSMA, FAP, vimentin, fibroblast-specific protein 1 (FSP-1/S100A4) and PDGF-Rα. However, these markers are neither exclusively, nor ubiquitously expressed by CAFs, thereby hampering further dissection of fibroblast subset function [18]. In contrast, immune subsets present in the TME of a variety of solid malignancies have been characterized extensively, which allows the analysis of the specific function of these subsets with regard to their role in anti-tumor immunity [34,35]. Therefore, recent efforts have focused on addressing this knowledge gap of fibroblast subset phenotype and function by mapping fibroblast heterogeneity through RNA-sequencing based approaches. Various research groups have tried to provide a roadmap of the different fibroblast subsets present in solid tumors, including pancreatic, colon, ovarian, breast, lung and head and neck cancer [36,37,38,39,40,41,42,43]. Interestingly, the number of fibroblast subsets identified varies between tumor types. Whether this reflects actual biological differences between CAFs present in different tumor types or is due to the different approaches used to map fibroblast heterogeneity (e.g., cultured primary fibroblasts vs. directly ex vivo isolated fibroblasts, bulk mRNA-seq vs single-cell mRNA-seq), remains to be determined.

Despite differences in technical approaches, multiple studies have identified a distinct CAF subset that is characterized by the high expression of pro-inflammatory cytokines, such as IL-6, Leukemia Inhibitory Factor (LIF), IL-1β, IL-11 and CXCL12 (Figure 1G). This particular fibroblast subset is detected in pancreatic, colorectal, ovarian, breast and lung cancer and is implicated in regulating the immune response, suggesting a shared biological mechanism amongst different solid malignancies [36,39,40,42,43]. Collectively, we will refer to this distinct fibroblast subset as ‘inflammatory-type cancer-associated fibroblasts (iCAFs)’ and, due to their high immunomodulatory gene expression, zoom in on this particular subset. The findings of studies addressing iCAF subset, are summarized in Table 1. However, despite their similar immunomodulatory gene expression, it is currently not known whether this similar iCAF phenotype shared between tumors can be further subdivided in multiple iCAF subsets with distinct functions.

A landmark paper, showing the pan-tumoral presence of iCAFs, demonstrated the existence of a CD10- and GPR77-positive iCAF subset in both breast- and lung cancer [44]. The presence of this iCAF subset, characterized by high expression of IL-6 and IL-8, is associated with poor prognosis in both tumor types. Similarly, in ovarian cancer the presence of an iCAF subset, represented by CD49e^+^FAPhigh CAFs, with high expression of IL-6 and CXCL12, is associated with dismal prognosis of patients, which is in accordance with findings in other solid malignancies [39,44]. In head and neck cancer, a PDGF-Rα+ CAF subset can be found that expresses high levels of CXCL12 [43]. Moreover, this study also finds a different fibroblast subset that expresses IL-6, suggesting that there might be additional heterogeneity within the iCAF subset. The immune regulatory role of these iCAFs has not been studied to date but, given the strong expression of key immune mediators, may provide novel insights into how CAFs modulate T-cell function.

Studies in a murine model of pancreatic ductal adenocarcinoma (PDAC) suggest that iCAFs upregulate pro-inflammatory cytokine expression through a paracrine tumor-fibroblast feedback loop [36]. Moreover, quiescent pancreatic fibroblasts, called pancreatic stellate cells, could be coaxed to adapt this inflammatory phenotype upon coculture with PDAC tumor organoids, showing the plasticity of fibroblasts in response to environmental cues. The exact, probably tumor-derived, factors that initiate this inflammatory transition remain to be elucidated, but this inflammatory state has been shown to be perpetuated by pro-inflammatory cytokines (IL-6, LIF) through both STAT3 and STAT4 dependent mechanisms [45,46]. The iCAF subset was also recently identified using single-cell RNA-sequencing of human PDAC derived CAFs [42].

In breast cancer, the CAF-S1 subset shows striking similarities with the iCAF subset found in pancreatic cancer with regard to inflammatory cytokine expression, including IL-1β, IL-6 and IL-17 [40]. Different histopathological breast cancer subsets (luminal, HER2^+^, triple-negative) show accumulation of iCAFs, albeit at different propensities. Strikingly, the highest percentage of iCAFS has been reported in a subset of triple-negative breast cancers, the breast cancer tumor type with the most dismal prognosis. Importantly, this study also revealed an association between the iCAF subset and composition of the immune infiltrate present in breast cancer, elegantly linking fibroblast subset to immune context. Triple-negative breast cancers with a high content of iCAFs showed higher infiltration of immune suppressive FOXP3^+^ T regulatory cells and a concomitant decrease in CD8^+^ T-cells [40]. In contrast to PDAC, where iCAFs were found distant from the malignant epithelial cells separated by a layer of myofibroblasts not exhibiting the inflammatory phenotype, the iCAFs in triple-negative breast cancer specimens were preferentially detected adjacent to malignant cells [36,40]. This disparity raises the question whether differential spatial distribution of iCAFs also influences T-cell activation and migration in the TME and is an important topic warranting further investigations.

Altogether, these studies show the first systematic assessment of CAF heterogeneity in the TME and provide novel insights into the functional role of different CAF subsets in a variety of solid malignancies. Furthermore, they point towards the iCAF subset as an important player to study with regard to T-cell activation and migration. In our opinion, future studies that incorporate and confirm the reported markers are crucial to reach a uniform definition of (i)CAFs. This will facilitate comparability between studies performed by different research groups and guide rational targeting of iCAFs to further enhance immunotherapeutic interventions.

### 2.2. Inflammatory Fibroblast Subsets in Autoimmunity

Immune regulation by fibroblasts also gathered attention in other fields of research, especially in the field of auto-immune diseases [47,48,49]. Interestingly, oncology and autoimmunity are two disease entities for which treatment has opposing goals, reinvigorating or suppressing T cell responses, respectively. Therefore, cross-pollination between oncology research and these disciplines could lead to a better understanding of fibroblast biology and their role in regulation of the T-cell response. Therefore, key findings with regard to fibroblast phenotypes in autoimmune diseases are summarized below.

Inflammatory bowel diseases (IBD) and rheumatoid arthritis (RA) are classical examples of autoimmune diseases in which autologous T-cells recognize self-peptides leading to colonic and synovial destruction, respectively. For both diseases, recent publications indicate a crucial role for fibroblasts in many aspects of T-cell function, including proliferation, migration and activation [50,51,52]. The immune regulatory fibroblast phenotype landscape is being unraveled in these diseases using (single-cell) RNA-sequencing, revealing the presence of fibroblast subsets, showing high similarities to iCAFs, that might drive inflammation and tissue damage [47,48,49].

In the first comprehensive study on the role of fibroblasts in the pathogenesis of ulcerative colitis (UC), Kinchen et al. showed the presence of five distinct fibroblast subsets along the crypt-villus axis of the normal human colon [49]. These subsets could also be found in colon biopsies from patients with UC. However, the S4 subset, characterized by the expression of TNFSF14 (LIGHT), IL-33 and Lysyl oxidases, was greatly expanded in the inflamed gut. Moreover, this subset also showed the highest IL-6 expression of all identified fibroblast subsets, showing similarity to the iCAF subsets found in tumors. The gene expression profile of the S4 subset revealed upregulation of genes involved in the regulation of leukocyte migration. Thus, the S4 subset identified in UC differs in this respect from the iCAF subset found in solid malignancies, where iCAFs are mostly associated with low T-cell infiltration. It would be of interest to directly compare the S4 subset found in UC with iCAF subsets in solid malignancies to further unravel this discrepancy.

Similarly, in RA distinct fibroblast subsets have been identified by independent research groups. Mizoguchi et al. showed the presence of three distinct fibroblast subsets in the human synovium based on the combinatorial expression of CD34, THY1 and CDH11 (CD34^−^THY1^−^CDH11^+^, CD34^−^THY1^+^ CDH11^+^, CD34^+^THY1^+^ CDH11^+^). The CD34^+^THY1^+^ CDH11^+^ fibroblasts displayed a cytokine expression profile closely resembling the iCAFs found in several solid malignancies, including high expression of IL-6, CXCL12 and CCL2. However, this subset was not expanded in RA and therefore less likely to contribute to the enhanced immune infiltration seen in this disease. In contrast, the CD34^−^THY1^+^CDH11^+^ subset, characterized by high CXCL12 expression but low IL-6 and CCL2 expression, was greatly expanded in RA, and mostly located in the perivascular zone in close association with clusters of lymphocytes. The presence of this subset also correlated with the number of synovium-infiltrating CD45^+^ immune cells and displayed high expression of TNFSF11, a molecule known to be involved in T-cell trafficking in autoimmune inflammation [53]. Further interrogation of the CD34^−^THY1^+^CHD11^+^ fibroblast secretome might reveal additional factors regulating leukocyte infiltration and lead to molecular insights to target this process. Recently, Croft et al. showed the presence of two distinct pathogenic, FAP^+^ fibroblast subsets, based also on the expression of THY-1. These cells were detected in the inflamed synovium in a murine model of serum transfer induced arthritis (STIA) and collagen-induced arthritis (CIA). THY1^+^ fibroblasts, displayed high expression of inflammatory cytokines, including IL-6, LIF, IL-33 and IL-34, resembling the CD34^+^THY1^+^CDH11^+^ subset, and were predominantly present in inflamed synovium. Interestingly, orthotopic transfer of the THY1^+^ fibroblasts, but not their THY1 negative counterparts, to inflamed ankle joints of CIA mice resulted in more severe and sustained joint swelling, including enhanced infiltration by CD4^+^ effector T-cells [48]. Direct comparison of these results with Mizoguchi et al. is difficult since the expression of CD34 on these fibroblasts was not reported. Therefore, it is not possible to distinguish between the effects on leukocyte infiltration of CD34^+^THY1^+^ and CD34^−^THY1^+^ fibroblasts, once again stressing the need for a standardized definition of fibroblast subsets. Finally, it would be of great interest to also study the presence and secretory profile of CD34^−^THY1^+^CHD11^+^ fibroblasts in solid malignancies and IBD to see whether the differences in leukocyte infiltration can be linked to this particular fibroblast subset.

These studies on fibroblast heterogeneity in inflammatory diseases do not only demonstrate the presence of inflammatory fibroblasts and their link to immune regulation, but also reveals the omnipresence of this inflammatory fibroblast phenotype in a wide range of diseases. However, it is also clear that different immunoregulatory fibroblast subsets appear to be present which display distinct immune regulating function with respect to immune cell infiltration. Direct comparison of these immune regulatory subsets, found in autoimmunity and oncology, might help to explain the association between the different types of fibroblasts and their opposing effects on the level of T-cell infiltration observed in these diseases. It may also provide clues to exploit the intratumoral fibroblast subpopulations, especially since the comparison of the stromal heterogeneity in normal and diseased tissues shows that the inflammatory state of these fibroblasts should be viewed as a dynamic response towards cues derived from their environment, rather than a static, fixed state. Therefore, it is also vital to understand the molecular mechanisms that induce the differentiation of fibroblasts towards these different immunoregulatory subsets.

## 3. Mechanisms of CAF-T-Cell Interactions

A firm link between the markers expressed by the different fibroblast subsets and their function will inevitably advance our understanding of fibroblast function. To date however, most studies have not incorporated phenotypical marker expression when reporting on functional studies of CAF–T-cell interactions, precluding the attribution of their findings to a marker-defined fibroblast subset. Therefore, we will discuss the literature regarding CAF–T-cell interactions independent of ‘marker-defined fibroblast subsets’ but rather describe CAFs based on their functional phenotype, i.e., their function with regard to the regulation of T-cell anti-tumor immunity.

### 3.1. Regulation of T-Cell Migration within the Desmoplastic TME

Migration of activated tumor-reactive T-cells towards the tumor is one of the crucial steps in an effective anti-tumor response and vital for the effectiveness of T-cell based immunotherapies. Broadly speaking, tumors can be divided in three immune phenotypes, based on the presence, exclusion or absence of T-cell infiltration [54,55,56,57]. Immune inflamed tumors are characterized by a high presence of intratumoral T-cells, indicating a (suboptimal) pre-existing antitumor T-cell response [58]. This immune phenotype is correlated to a better clinical outcome in a variety of solid malignancies [59,60,61]. Moreover, the immune phenotype also appears to be a good predictor for patient survival [62]. Finally, the immune inflamed phenotype is also associated with better responsiveness to current immunotherapeutic interventions [63,64]. In contrast, the immune excluded and immune deserted phenotypes are characterized by exclusion or absence of T-cell infiltration in the tumor cell bed, respectively. These latter two phenotypes are associated with a lack of responsiveness to immune checkpoint blockade [65]. We will discuss the role of CAFs on T-cell migration based on the conceptual framework provided by the tumor immune phenotype classification and address the contribution of CAFs to this phenotype through their secretome and modulation of the extracellular matrix.

#### 3.1.1. The CAF Secretome

T-cell trafficking relies on cytokine-chemokine morphogenic gradients that guide peripheral T-cells towards the tumor. CAFs, upon their activation in the TME, produce an abundance of these cytokines and chemokines thereby enabling them to regulate T-cell migration. Different factors secreted in the TME drive CAF activation, including members of the TGF-β superfamily, PDGFs, epidermal growth factors (EGFs), sonic hedgehog (SHH) and interleukins [9,16,19,66]. The resulting activated CAFs produce a plethora of cytokines and chemokines, including IL-6, IL-8, IL-10, tumor necrosis factor α (TNFa), CCL2, CCL5, CXCL9, CXCL10 and TGF-β [9,67,68]. It is not well understood which of these factors are key in regulating T-cell migration in vivo. However, recently, two independent studies have shown the role of TGF-β signaling in CAFs on T-cell infiltration and the subsequent response to checkpoint inhibition [69,70]. In a genetically reconstituted murine model of CRC, Tauriello et al. showed an increased stromal TGF-β signature along the CRC normal mucosa-adenoma-carcinoma mutational sequence. Moreover, this increase in a TGF-β induced gene program in CAFs coincided with the exclusion of T-cells from invasive cancers, closely resembling the immune exclusion phenotype seen in a subset of CRC patients [59]. In accordance, monotherapy with PD-1/PD-L1 blockade did not result in anti-tumor activity in mice with CRC-derived experimental liver metastasis. However, prior treatment with galunisertib, a TGFβR1 kinase inhibitor, led to increased recruitment of CD3^+^ T-cells to CRC-colonized livers and reduced overall metastatic burden. Combined targeting of both the TGF-β and PD-1/PD-L1 axis showed synergistic effects, by facilitating T-cell infiltration and potentiation of tumor-reactive T-cells, leading to complete tumor eradication in the majority of treated animals. Similarly, in patients with metastatic urothelial cancer, Mariathasan et al. showed that therapeutic failure of PD-L1 blockade corresponded to an activated TGF-β signaling signature in CAFs. Patients with high stromal TGF-β signaling showed exclusion of CD8^+^ T-cells from the tumor parenchyma and responded poorly to checkpoint inhibition. This was confirmed in a murine breast cancer model (EMT 6) that recapitulated the immune excluded phenotype. Combined targeting of the TGF-β and PD-1/PD-L1 axis in this model also led to increased T-cell infiltration and a vigorous anti-tumor response, in concordance with previous findings [70,71]. Of note, the downstream target genes induced by TGF-β signaling in CAFs and their contribution to the observed T-cell exclusion in these tumors have not been elucidated but will provide further insights into the underlying mechanisms of CAF-mediated T-cell exclusion.

#### 3.1.2. Extracellular Matrix Production (ECM) and CAF Barrier Function

Another factor inhibiting the ability of T-cells to reach the tumor parenchyma is the physical barrier that CAFs impose around the tumor. During quiescence, resting fibroblasts are hardly synthetically active and reside within the physiological extracellular matrix. Upon activation, CAFs strongly increase their ability to actively remodel the ECM. Intriguingly, ECM production is also a largely TGF-β driven process, supporting the hypothesis that TGF-β is a cytokine with a multifaceted role in immunosuppression [72,73,74]. The net result of the ECM remodeling process is the formation of a dense extracellular matrix network surrounding the tumor epithelium, while maintaining a relatively loose density within stromal regions. In human lung and colorectal tumors, this altered deposition of ECM determines the capacity of T-cell migration, restricting access to the densely encapsulated tumor epithelium [75,76]. In lung cancer, this immune excluded phenotype could be partly restored by remodeling of the tumor-surrounding ECM by treatment with collagenase D. Similarly, chemically targeting the ECM production by CAFs in murine models of lung cancer and melanoma also resulted in enhanced CD8^+^ infiltration [77]. In addition, direct targeting of FAP^+^ CAFs, was shown to increase recruitment of CD8^+^ T-cells and subsequent tumor control in murine models of lung, colon and pancreatic cancer. This was most likely the result of both decreased ECM production as well as decreased T-cell retention in stroma due to secreted factors by these CAFs [78,79].

Altogether, CAFs can modulate T-cell access to the tumor both through inhibition of T-cell migration through a TGFβ-dependent gene program, as well as by altering the composition of the ECM. As such, CAFs appear to be pivotal in the T-cell immune exclusion phenotype observed in a variety of solid malignancies.

### 3.2. CAF-Mediated T-Cell Suppression

The immune inflamed and excluded phenotype are both characterized by the presence of T-cells suggesting the existence of a pre-existing anti-tumor response, albeit that in the latter phenotype the T-cells are retained in the tumor stroma and barely migrate to the tumor bed. However, in both cases T-cell effector function is often suppressed [58]. Different factors mediate T-cell suppression, ranging from inadequate co-stimulation by professional antigen-presenting cells (APC) to direct suppression through immune checkpoint inhibition [33]. We will discuss the immune suppressive role of CAFs on T-cells through (1) modulation of antigen presentation and (2) regulation of checkpoint molecule expression.

#### 3.2.1. Modulation of Antigen Presentation

Tumor antigen uptake by professional APCs, in particular dendritic cells (DC), at the tumor site and subsequent trafficking to the lymph nodes followed by activation and co-stimulation of antigen-specific T-cells, is the crucial first step in generating an adaptive, T-cell mediated anti-tumor response. Therefore, DCs are a prime target for CAF-mediated T-cell suppression. The enhanced secretion of multiple factors including TGF-β1 by activated CAFs, can potentially interfere with efficient DC trafficking [16]. Indeed, TGF-β1 impairs the migratory ability of DCs and thereby adequate T-cell activation in the tumor-draining lymph nodes [80]. Moreover, TGF-β1 has been shown to directly decrease the expression of important co-stimulatory molecules (e.g., CD80, CD86), thereby promoting an immature DC phenotype that is less well equipped to fully potentiate tumoricidal T-cells [81]. These so called tolerogenic DCs have been implicated to promote T-cell differentiation towards a regulatory phenotype, further impairing effective anti-tumor immunity [82].

Next to TGF-β, CAF-derived IL-6 has also been shown to prevent the maturation of DCs through the IL-6-gp130-STAT3 axis [83,84]. By altering the expression of key factors involved in proper T-cell activation, DCs are able to drive or halt T-cell expansion. In hepatocellular carcinomas it has been shown that CAFs induced IDO-producing, regulatory DCs, thereby altering the metabolic milieu in which the tumor-reactive T-cells reside, resulting in a decreased anti-tumor response [84]. By interfering with multiple aspects of DC biology, CAFs are able to prevent proper activation of tumor-reactive T-cells.

#### 3.2.2. Checkpoint Molecule Expression

T-cell dysfunction is characterized by a (partial) loss of effector function and proliferative ability due to the activation of different inhibitory pathways, the so-called immune checkpoints, which are highly expressed on chronically activated tumor-reactive T-cells [85,86,87,88,89,90]. Underlying this dysfunctional state is the persistent T-cell receptor activation encountered by these cells in the TME [91]. Expression of different immune checkpoint receptors on intratumoral T-cells makes them susceptible to inhibition via ligand-receptor signaling. Fibroblasts express multiple of the known immune checkpoint ligands, including PD-L1, PD-L2, galectins and the enzyme IDO [52,92,93,94]. The functional consequences of checkpoint molecule expression on CAFs is slowly being unraveled. Nazareth et al. demonstrated both activating and suppressing activities of CAFs in co-culture with T-cells. Fibroblast-induced T-cell suppression could be alleviated by blockade of either PD-L1 or PD-L2 on CAFs [95]. Importantly, PD-L1 expression has been shown to be upregulated on fibroblasts cultured in vitro [96,97]. Therefore, the functional role of immune checkpoint molecules on CAFs should be preferentially investigated in in vivo genetic perturbation models.

Next to immune checkpoint expression, CAFs also play an indirect role in checkpoint-mediated T-cell suppression by secreting factors which upregulate checkpoint molecules on other cell types in the TME. For instance, tumor cells and neutrophils both upregulate PD-L1 under the influence of CAF-secreted factors [98,99]. Intriguingly, different CAF-secreted factors were responsible for the upregulation of PD-L1 in different cell types. CXCL5 induced the expression of PD-L1 in both murine melanoma and colon cancer cell lines through activation of the CXCR2 receptor [98]. In contrast, CAF-derived IL-6 induced PD-L1 expression in neutrophils, through STAT3 dependent mechanisms, in human hepatocellular carcinoma [99]. Therefore, by regulating the expression of checkpoint molecules on different cell types present in the TME, CAFs are able to inhibit T-cell expansion indirectly, hampering an effective anti-tumor response. Further interrogation of the CAF secretome might reveal other factors that are able to induce checkpoint molecule expression on cells present in the TME.

In conclusion, CAFs actively regulate both the migratory and activation status of T-cells in the TME. By exclusion of T-cells, either through secreted factors or by generating a physical ECM barrier, CAFs are actively involved in shaping the immune infiltrate in solid malignancies. In addition, T-cells that are able to reach the tumor may be functionally inhibited by CAFs, either through endogenous expression of checkpoint molecule ligands or by upregulation of these molecules on other cells in the TME. Knowledge of the mechanisms employed by these CAFs, might also lead to insights in potential therapeutic targets that could pave the way for further enhancement of T-cell based immunotherapy.

## 4. Therapeutic Targeting of the CAF–T-Cell Axis in Solid Malignancies

The deeper understanding of CAF heterogeneity and their role in T-cell biology paves the way for selective targeting of immunosuppressive fibroblast subsets and their immunosuppressive mechanisms. Further improvement of current immunotherapeutic strategies is expected when they are combined with regimens targeting different aspects of the immunosuppressive TME [100]. Several strategies to either (1) directly or (2) indirectly target CAFs or CAF-associated immunosuppressive mechanisms will be discussed here and are summarized in Figure 2.

### 4.1. Direct CAF Targeting

Targeting of distinct CAF subsets by targeting of their specific antigens, is an attractive option to selectively deplete immunosuppressive CAFs from the TME (Figure 2A). Successful application of such therapies relies on CAF-selective expression of the targeted antigen in order to circumvent ‘on-target off-CAF’ toxicity. The most well-studied example of CAF targeting is FAP. In two different studies, using either genetic FAP ablation or a DNA vaccine directed against FAP, improved recruitment of CD8^+^ T-cells and prolonged survival in murine models of lung, colon and pancreatic cancer [78,79]. However, subsequent experiments of independent research groups using FAP-reactive, CAR transduced T-cells or FAP ablation using diphtheria toxin, also showed that these interventions also could target both murine and human bone marrow stromal cells (BMSCs) [101,102]. In these murine models, FAP depletion led to lethal bone toxicity and cachexia, indicating the potential risks of direct CAF targeting. However, FAP-reactive CAR T cells have now also been employed without significant toxicity and with the ability to enhance anti-tumor immunity in murine models of lung, colon and pancreatic cancer [103,104,105]. The disparity between the toxicities found in these studies is probably due to the differences in CAR constructs used. A first in-man study in malignant pleural mesothelioma demonstrated good tolerability of FAP CAR T cells [106]. Targeting of FAP with a humanized antibody, sibrotuzumab, in patients with metastatic CRC also demonstrated good tolerability but did not result in tumor control, indicating the presence of dominant alternative mechanisms that hamper an effective anti-tumor response, once again stressing the need for a combinatorial approach to obtain an effective cancer treatment [107].

Another warning stems from the observation that targeting of Sonic hedgehog (Shh) and αSMA in CAFs led to acceleration of tumor progression in murine models of PDAC, including an increase in CD4^+^FOXP3^+^ T regulatory cells [30,31]. This emphasizes the need for identification of fibroblast subset markers and function to be able to target pathogenic fibroblast subsets. Therefore, more specific and CAF-exclusive antigens, as identified through RNA-sequencing based approaches may lead to better markers that enable the selective targeting of immunosuppressive CAFs. In this regard, direct targeting of iCAF subsets seems like a potential avenue to manipulate the immune infiltration and activation in the TME. Finally, human clinical trials should carefully monitor the subsequent immune response followed by the administration of therapies targeting CAFs to identify and potentially synergize different treatment strategies in order to optimize anti-tumor responses.

### 4.2. Indirect CAF targeting

#### 4.2.1. Targeting the CAF Secretome/CAF Signaling

Currently, direct targeting of iCAFs is not feasible due to the absence of a specific marker that overcomes the ‘on-target off-CAF’ toxicity observed in previously tested drugs. Therefore, targeting of the CAF-derived factors involved in T-cell exclusion and suppression represents an alternative approach (Figure 2B). Due to the important role of TGF-β signaling in the immune excluded phenotype, and the improved T-cell migration attained upon blockade of TGF-β signaling in murine models of cancer, this signaling pathway represents an attractive option for therapeutic intervention [69,70]. While the first generation TGF-β inhibitors were associated with (severe) side effects in preclinical models, including cardiac toxicity [108,109], the next-generation TGF-β kinase inhibitors, antisense oligodeoxynucleotides and neutralizing antibodies that bind TGF-β directly show good tolerability in ongoing clinical trials [110,111,112]. Combinatorial targeting of the TGF-β and PD-1/PD-L1 axis offers the potential to improve both T-cell infiltration as well as alleviation of PD-1-mediated T-cell suppression and is especially attractive to treat malignancies notorious for a high stroma content and immune excluded phenotype. This strategy is currently being tested in a variety of solid malignancies, including glioblastoma, lung, hepatocellular and pancreatic cancer [113]. Alternative molecules implicated in T-cell suppression by CAFs, both through direct and indirect mechanisms, are IL-6 and CXCL12 [83,84,114]. Monoclonal antibodies targeting IL-6 are currently approved for use in the treatment of autoimmune diseases and are also used to suppress the life-threatening cytokine release syndrome which can occur after CAR T-cell therapy [115]. These drugs could be repurposed for alleviation of (iCAF-derived) IL-6 mediated immunosuppression in solid malignancies, which has been demonstrated to act synergistically with PD-L1 blockade in a murine model of melanoma [116]. Next to IL-6, the safety of CXCL12/CXCR4 antagonists is currently being tested in multiple solid malignancies and might provide an alternative route to improve T-cell infiltration in the TME [117]. Further insights into the immunosuppressing factors produced by CAFs could identify novel signaling routes that can be modulated to improve T-cell based immunotherapy.

#### 4.2.2. Modulation of the Extracellular Matrix

Since tumor infiltration by T-cells is key for an effective anti-tumor response, and modulation of the ECM by CAFs prevents T-cell infiltration, it provides a potential target for improving T-cell based immunotherapy [75]. In this regard, repurposing of drugs currently used for the treatment of fibrotic diseases (e.g., pulmonary and hepatic fibrosis) represents an attractive approach that could be readily translated to the clinic. Pirfenidone, an antifibrotic drug that modulates the production and deposition of ECM by CAFs by both inhibition of CAF proliferation and inhibiting TGF-β mediated stimulation of ECM production, is a prime example of an antifibrotic drug that, theoretically, could be used to stimulate T-cell infiltration [118,119]. Preclinical studies showed a reduction of collagen and hyaluronan levels in murine models of breast cancer after pirfenidone treatment [120]. Intriguingly, the risk of lung cancer appears to be lower in patients with pulmonary fibrosis treated with pirfenidone, possibly due to alleviation of immunosuppression in the TME [121]. Other therapies that can be used to target the ECM are SMO inhibitors, collagenase, hyaluronidase, anti-fibronectin, and anti-TN-C monoclonal antibodies, which have already been discussed elsewhere [122]. The respective roles of these drugs on enhancing immune infiltration is currently not understood and should be investigated in further studies.

## 5. Conclusions

In order to improve current T-cell based immunotherapeutic interventions by employing anti-CAF therapies, several hurdles should be tackled. First and foremost, fibroblast subsets present in the TME should be characterized extensively to enable the study of immune modulation by specific (i) CAF subsets and identify (i) CAF-selective targets. Therefore, current subset identification efforts through RNA sequencing should be extended and verified in different tumor types. Ultimately, consensus on the nomenclature and used markers for the identification of the different CAF subsets should be reached to facilitate comparability of studies between research groups and guide rational targeting.

Although the research into CAF subsets and their immunomodulatory properties is in its infancy, they appear to be potent modulators of the immune system and therefore an attractive target for improving T-cell based immunotherapy. Specifically, the iCAF subset, characterized by the expression of inflammatory cytokines (TGF-β, IL6, CXCL12), and present in a variety of solid malignancies appears to be involved in shaping the immune landscape in these tumors. However, the exact influence of this iCAF subset on T-cell function remains to be elucidated. In our opinion, the striking resemblances in gene expression in iCAFs present in different tumor types should be studied on the functional level to investigate if these iCAFs regulate T-cell function in a similar way or whether there are tumor type-specific differences. Moreover, clues to the divergent functions in immune regulation within the iCAF subset are starting to be discovered by comparison of two immune phenotypes at the opposites ends of the spectrum, i.e., immune regulation in autoimmunity and oncology. Within the autoimmunity field, distinct fibroblast subsets have now been identified that are correlated to enhanced lymphocyte infiltration. In contrast, many tumors are characterized by the exclusion of T cells, hindering the successful application of T-cell based immunotherapy. Therefore, the differences between inflammatory fibroblasts in autoimmunity and iCAFs in oncology should be investigated in more detail to unravel mechanisms for improving T-cell infiltration within the TME. In this respect, it is important to note that the inflammatory state of fibroblasts appears to be plastic and thus should be viewed as a dynamic response towards TME derived signals. Deeper understanding of the signaling pathways involved in this plasticity could ultimately lead to the possibility of skewing CAFs towards an immune promoting phenotype.

Our incomplete understanding of the identified CAF subsets and their function hinders the direct targeting of CAFs at the current stage without risking ‘on-target off-CAF’ toxicity, although there are encouraging results that show the possibility of targeting the stroma (e.g., FAP targeting with CAR T-cells). Therefore, indirect targeting of the CAF-derived factors involved in T-cell exclusion and suppression currently represents a valid option in complementing current T-cell-based immunotherapies. T-cell exclusion is facilitated by CAFs through modulation of the ECM and is a process highly dependent on active TGF-β signaling. Blockade of the TGF-β pathway represents an attractive option to facilitate T-cell invasion into the tumor. Finally, T-cell suppression by CAFs within the TME can be the result of inhibitory checkpoint molecules expressed on CAFs themselves or the result of CAF-mediated suppression by other cell types present in the TME. Altogether, the goal of indirect CAF targeting is thus twofold: (1) Facilitate trafficking of tumor-specific T-cells to the tumor bed by interfering with the barrier function of CAFs and (2) release the immune suppression imposed by CAFs on these tumor-reactive T-cells.

In conclusion, the knowledge in the field of CAF biology is rapidly increasing and showing multiple pathways via which CAFs are able to interfere with an effective tumor-specific T-cell response. This knowledge is now gradually translating to concrete interventions that in the future can hopefully be incorporated in T-cell based immunotherapies to broaden their use and effectivity as treatment for patients with cancer.

*Article* *highlights*
RNA sequencing has revealed the presence of multiple subsets of cancer-associated fibroblasts (CAFs) in multiple types of solid malignanciesThe inflammatory CAF (iCAF) subset, characterized by the expression of inflammatory cytokines (TGF-β, IL6, CXCL12), is present in a variety of solid malignancies and may shape the tumor immune microenvironmentThe inflammatory state of iCAFs should be viewed as a dynamic response towards cues within the tumor microenvironment, rather than a static, fixed stateT-cell excluded tumors are strongly associated with Transforming Growth Factor-β (TGF-β) signaling in CAFsCAF-mediated regulation of APC function and T cell checkpoint inhibition contributes to the exhausted T cell phenotypeTargeting of the CAF secretome, TGF-β signaling in CAFs and its ECM-producing functions may improve T cell infiltration and, in combination with checkpoint inhibitors, lead to a release of immune suppression in the TMEThe knowledge gap regarding specific CAF subset function limits the exploitation of direct CAF targeting without significant off-target risks


## Figures and Tables

**Figure 1 jcm-08-01989-f001:**
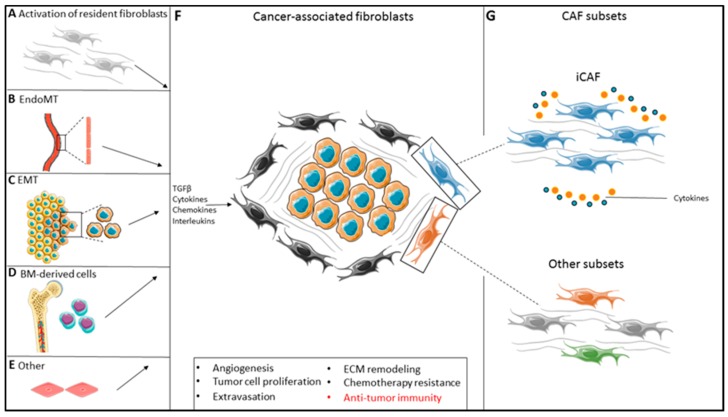
Fibroblast heterogeneity in the tumor-microenvironment. (**A**–**E**). The origin of CAFs in the TME is diverse and they can be either derived from the activation of resident fibroblasts (**A**), endothelial-to-mesenchymal transition (EndoMT) (**B**), epithelial-to-mesenchymal transition (EMT) (**C**) bone-marrow derived mesenchymal cells (**D**) and/or other differential pathways (e.g., smooth muscle cell trans-differentiation (**E**)). (**F**,**G**). The function of these CAFs is diverse (**F**) and regulated by cues derived from within the TME, leading to formation of subsets with specific functions, including but not limited to, iCAFs (**G**). TGF-β, transforming growth factor β; ECM, extracellular matrix; CAF, cancer-associated fibroblast; iCAF, inflammatory CAF.

**Figure 2 jcm-08-01989-f002:**
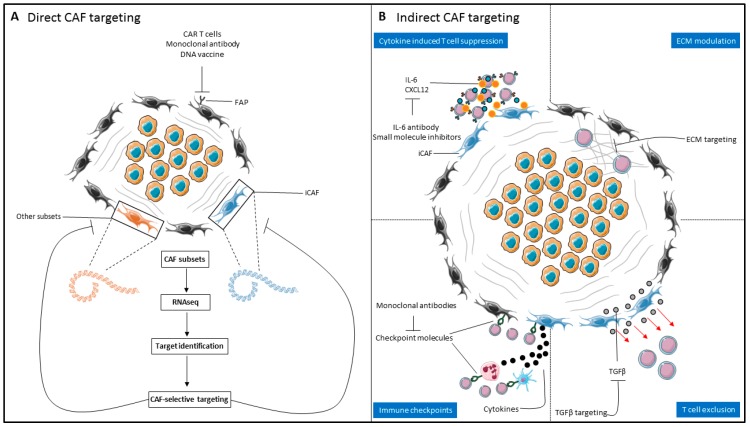
Therapeutic targeting of CAFs to enhance T-cell based immunotherapies. (**A**) Direct CAF targeting relies on the identification of CAF-selective targets to reduce ‘on-target off-CAF’ toxicity. RNA-sequencing based approaches enable identification of markers to target specific immunomodulatory CAF subsets. (**B**) Indirect CAF targeting relies on targeting of CAF-derived factors involved in T-cell exclusion and suppression. This can be done either through inhibiting CAF-derived cytokines involved in suppressing T-cell function (top-left), modulation of the ECM (top-right) or blocking of inhibitory chemotactic signals (e.g., TGF-β signaling) to improve accessibility of T-cells to the tumor (bottom-right) and/or inhibition of checkpoint molecules on CAFs or associated immune cells to potentiate tumoricidal T-cell effector functions (bottom-left). CAR, chimeric antigen receptor; FAP, fibroblast activation protein; CAF, cancer-associated fibroblast; iCAF, inflammatory CAF; IL-6, interleukin-6; ECM, extracellular matrix; TGF-β, transforming growth factor β.

**Table 1 jcm-08-01989-t001:** Inflammatory fibroblast subset characteristics.

Tumor/Disease Type	Fibroblast Subsets	Human/Mouse	iCAF Subset Characteristics	Reference
PDAC	2	Mouse	Low αSMA expression, high IL-6, IL-11, LIF expression.	Öhlund D. et al. [36]
Ovarian cancer	2	Human	FAP^+^, high IL-6 and CXCL12 expression.	Hussain A. et al. [39]
Breast cancer	4	Human	CAF-S1 subset, high IL-1β, IL-6, CXCL12 and IL-17 expression. Induces differentiation of Tregs.	Costa A. et al. [40]
PDAC	3	Human and mouse	Low αSMA expression, high IL-6, IL-8, CXCL1, CXCL2, CXCL12 expression.	Elyada E. et al. [42]
Breast and lung cancer	N/a	Human	CD10^+^GPR77^+^, high IL-6 and IL-8 expression.	Su S. et al. [44]
Head and neck cancer	4	Human	1: PDGF-Rα+, high CXCL12 expression. 2: High αSMA expression, high IL-6 expression.	Puram S. et al. [43]
Rheumatoid arthritis	3	Human	CD34^−^THY1^+^CDH11^+^: high IL-6 and TNFSF11 expression. Greatly expanded (compared to normal synovium) and associated with perivascular lymphocyte clusters in RA.	Mizoguchi F. et al. [47]
Rheumatoid arthritis	2	Mouse	THY1^+^ fibroblast, high expression of IL-6, LIF, IL-33 and IL-34. Linked to enhanced CD4^+^ T-cell infiltration.	Croft A. et al. [48]
Ulcerative colitis	5	Human	CAF-S4 subset, high IL-6, TNFS14, IL-33 and lysyl oxidases expression. Greatly expanded (compared to normal colon) in UC.	Kinchen J. et al. [49]

IL, interleukin; CXCL, C-X-C motif chemokine; LIF, leukemia inhibitory factor; FAP, fibroblast activation protein; PDAC, pancreatic ductal adenocarcinoma; CRC, colorectal cancer; NSCLC, non-small-cell lung carcinoma; UC, ulcerative colitis; RA, rheumatoid arthritis.

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
