# Peer review of "Targeting of the Cancer-Associated Fibroblast—T-Cell Axis in Solid Malignancies"

_jcm, 2019, doi:10.3390/jcm8111989_

Round 1

Reviewer 1 Report

The authors present a comprehensive review about the current knowledge on CAFs and their interaction with tumor infiltrating T-cells as well as possible therapeutic interventions. Overall, the review is very well written and there are only a few points to be addressed.

Page 1, line 15: most tumors do not express tumor-specific antigens, but rather tumor-associated ones. Maybe rephrase to: "despite the fact that their tumors are recognized by the immune system".

Page 1, line 30: A reference supporting the statement should be added.

Page 2, line 63: Figure 1F

Page 2, line 72: vaccination does not activate tumor-reactive T-cells, but the rationale is to prime new T-cells.

Page 2, line 73: Radiotherapy and chemotherapy are not "immunotherapy" and do much more than activation of T-cells (there foremost goal is to kill tumor cells, immune effects include immunogenic cell death, DC maturation, etc)

Table 1 and chapter on autoimmune diseases: the authors should make clear, that overall the therapeutic goal in cancer and autoimmune diseases concerning T-cells is an opposite one. In cancer Th1 polarized T-cell response and cytotoxic T cells are wanted, whereas in autoimmune diseases these are the major players in pathogenesis. However, I completely agree that the two fields can learn a lot from each other. Having the same CAFs in both disease types poses questions about how the same CAFs can drive cancer on the one hand and autoimmune phenomena on the other (being on the opposite spectrum as stated by the authors themselves in the discussion section).

Page 7, line 216: As far as I understood the data, it is mostly association data and not studying the effect of CAFs on T cells.

Page 8, lines 239-242: The cited data refer to immune checkpoint inhibition (there are much more "current immunotherapies"). Please rephrase.

Page 11, line 371: Having the targeted antigen on normal cells leads to on-target toxicity compared to off-target toxicity with a low-specificity small molecule or antibody.

Figure 2B: lower right: rephrase to "immune checkpoints" (in all other panels the targeted mechanism / structure is stated and not the intervention)

Figure 2B: TGFbeta is not the only factor responsible for T-cell-exclusion. This should at least be mentioned in the legend.

Page 13, line 432: MMP inhibitors should lead to a more compact ECM. The rationale to use them is that cancer cells use MMPs for invasion and metastasis formation.

There seem to be some double-spaces.

Please make sure to superscript "+" for pos. stained cells.

Please put "in vivo" etc in italics.

Reviewer 2 Report

This manuscript provides a comprehensive review of the current literature on cancer associated fibroblasts and their relationship with the T cell immune axis. It has been well structure and provides a good overview.

Minor points:

please clarify lines 54-56: Is the author implying here that the ability of endothelial and epithelial cells to transition to a mesenchymal phenotype, via EndoMT - there by transforms these cells in to CAFs/ CAF phenotype?

Line 63-64: This is a new point, unrelated to CAF heterogeneity, and could be moved into the next paragraph which discusses the role of CAFs in the regulation of anti-tumour immunity.

Reviewer 3 Report

The paper is well written, however it lacks clarity and depth in certain areas. The authors should made substantial improvements to the text, expand on the discussion and add more references, before it is suitable for publication.

- Figure 1 should be improved. Illustration and listing of CAF subsets should be expanded to include more than just iCAFs and "other subsets." The authors should be more specific and thorough. 

- Lines 101-102: The authors should make a clearer distinction between the various CAF subsets. For instance, in Table 1 they list iCAFs but in the text they do not specifically refer to the table just representing iCAFs. This is a bit confusing and will be to readers.

- The authors should discuss findings, as they related to CAFs, from Puram, S. et al. (2017) Cell 171, 1611–1624.e24.

- The manuscript does not discuss, nor mention, the effect of CAFs on tumor-infiltrating lyphocytes, of which T cells are significant constituents

- Line 377-378: The statement that a FAP-CAR is toxic so, as such, is unsuitable for clinical trials is misleading, especially as it seems to have been made based on a single study from 2013. Authors should expand on this, including discuss the more comprehensively FAP-CAR-T cell studies published since. They should provide more evidence supporting their claim. There are also FAP clinical trials that have been underway since then.     

- Line 411: IL-6 anti-therapy is currently already implemented in the context of adoptive transfer T cell therapy. This needs to be considered
